Long-term follow-up of applying autologous bone grafts for reconstructing tympanomastoid defects in functional cholesteatoma surgery

Lan Wei-Che 1
Wang Ching-Yuan 1
Tsai Ming-Hsui 1
Lin Chia-Der d18546@mail.cmuh.org.tw 1 2
1 Department of Otolaryngology Head and Neck Surgery, China Medical University Hospital , Taichung , Taiwan
2 School of Medicine, China Medical University , Taichung , Taiwan
Andaloro Claudio
Electronic publication date: 2021 Nov 23
Publication date: 2021
Volume: 9
Electronic Location ID: e12522
Received 2021 Aug 17; Accepted 2021 Oct 29
Copyright: ©2021 Lan et al.
Copyright year: 2021
Copyright holder: Lan et al.
License: This is an open access article distributed under the terms of the Creative Commons Attribution License, which permits unrestricted use, distribution, reproduction and adaptation in any medium and for any purpose provided that it is properly attributed. For attribution, the original author(s), title, publication source (PeerJ) and either DOI or URL of the article must be cited.
License URL: https://creativecommons.org/licenses/by/4.0/

Keywords: Cholesteatoma, Surgery, Hearing, Prognosis, Long-term

Funding: China Medical University DMR-108-193 This work was supported by China Medical University (DMR-108-193). The funders had no role in study design, data collection and analysis, decision to publish, or preparation of the manuscript.

==============================
Objectives

This study investigated the long-term surgical outcomes of functional cholesteatoma surgery with canal wall reconstruction using autologous bone grafts as the primary material in patients with acquired cholesteatoma.

Subjects and Methods

Medical charts were retrospectively reviewed for all patients admitted to one institution for surgical intervention between 2010 and 2018. We analyzed 66 patients (66 ears) who underwent functional tympanomastoidectomy involving the use of autologous bone grafts for canal wall defect reconstruction. Surgical outcomes were evaluated by comparing preoperative audiometric results with follow-up data (at least 36 months after surgery). Logistic regression analyses were performed to determine prognostic factors related to long-term hearing success. These factors included classification and stage of cholesteatoma, stapes condition, ossicular chain damage, active infection of the middle ear, state of the contralateral ear, preoperative hearing thresholds, gender, and age.

Results

The mean follow-up period was 49.2 months. The recidivism rate was 6% (four of 66 ears). The pure-tone average significantly improved from 50.78 ± 19.98 to 40.81 ± 21.22 dB hearing level (HL; p < 0.001). Air–bone gaps significantly improved from 26.26 ± 10.53 to 17.58 ± 8.21 dB HL (p < 0.001). In multivariate logistic regression analysis, early-stage disease (p = 0.021) and pars flaccida cholesteatoma (p = 0.036) exhibited statistically significant correlations with successful hearing preservation.

Conclusion

Functional cholesteatoma surgery with autologous bone grafts reconstruction is an effective approach to significantly improve hearing with low recidivism rates. Localized disease and pars flaccida cholesteatoma were two independent predictors of successful hearing preservation.

Introduction

Surgical removal of cholesteatoma is the standard treatment once the diagnosis is established (Kuo, Liao & Shiao, 2015). However, cholesteatoma surgery is particularly challenging because the goals of removing the complete cholesteatoma and maintaining the normal functions of the temporal bone appear to conflict. Canal wall down (CWD) and canal wall up (CWU) mastoidectomies are the two main surgical techniques conventionally used for the management of middle ear cholesteatoma (Hellingman et al., 2019). Compared with CWD mastoidectomy, CWU mastoidectomy results in more favorable hearing outcomes and prevents cavity problems (Lucidi et al., 2019). However, studies have reported higher rates of recidivism that may be attributable to insufficient exposure in CWU mastoidectomy (Lucidi et al., 2019; Chadha et al., 2006; Kerckhoffs et al., 2016). A functional cholesteatoma surgical technique combining the advantages of both CWU and CWD mastoidectomies has been developed. Different names have been used for this developed technique including flexible endaural approach (Alleva et al., 1989; Paparella & Jung, 1984), tailor-made mastoidectomy (Kuo, Liao & CF, 2012), retrograde mastoidectomy (Dornhoffer, 2000; Minovi et al., 2014; Shewel & Abougabal, 2020), the inside-out approach (Roth & Haeusler, 2008), and limited mastoidectomy (Nikolopoulos & Gerbesiotis, 2009). Although differences exist among these various procedures, their main principle remains the same—tracking lesions to their ends. In this procedure, uninvolved middle ear cleft structures and the ear’s normal function are preserved. Acceptable recidivism rates have been reported for functional mastoidectomy (Hatano, Ito & Yoshizaki, 2010; Dornhoffer, 2004) compared with CWU and CWD mastoidectomies (Tomlin et al., 2013).

Mastoid obliteration and canal wall reconstruction, followed by disease removal, are mandatory to prevent leaving a cavity (Harun et al., 2015). Various materials have been used for mastoid obliteration including a bone plate (Shewel & Abougabal, 2020), aural cartilage (Dornhoffer, 2004), and fat (Lee et al., 2005). In this study, we used autologous bone grafts as the primary material for reconstruction. Studies examining the long-term follow-up outcomes of the application of autologous bone grafts for reconstructing tympanomastoid defects in functional cholesteatoma surgery are lacking. Therefore, the present study evaluated long-term hearing outcomes of functional mastoidectomy in which autologous bone grafts were used as the primary material for reconstruction. In addition, this study analyzed factors that can predict successful hearing preservation.

Materials and Methods

This retrospective study evaluated patients with acquired middle ear cholesteatoma who were admitted for functional tympanomastoidectomy in a single tertiary hospital between 2010 and 2018 with a minimum follow-up of 3 years. A total of 116 patients were assessed for eligibility, and of these 116 patients, 50 were excluded. Finally, 66 patients (66 ears) aged between 18 and 74 years were enrolled in this study. All of the surgeries were performed by a single surgeon. Exclusion criteria were as follows: (1) unavailability of complete audiometry data or follow-up hearing examination data for less than 3 years postoperatively, (2) revised surgery, (3) presence of congenital cholesteatoma, (4) diagnosis other than cholesteatoma, (5) undergoing a procedure other than functional tympanomastoidectomy, and (6) cholesteatoma of the external auditory canal. The study flow diagram is shown in Fig. 1. This study was approved by the Institutional Review Board of China Medical University Hospital (project approval number CMUH110-REC3-004). All methods were carried out in accordance with relevant guidelines and regulations. This study involved no prospectively collected data so there was no access to patients or opportunity to seek informed consent. Consent for waiver of consent from Institutional Review Board was obtained.

Figure 1 The study flow diagram.

The following detailed information was collected for each patient: age, sex, affected ear, follow-up duration, types of tympanoplasty, group and stage of cholesteatoma, severity of ossicle destruction, presence of the stapes suprastructure, state of the contralateral ear, and infection at the time of surgery (Table 1). The group and stage of cholesteatoma were determined according to the classification and staging system proposed by the European Academy of Otology and Neurotology/Japan Otological Society (EAONO/JOS) Joint Committee (Yung et al., 2017). In this study, patients were classified into two main groups: (1) pars flaccida and (2) pars tensa or combination of pars flaccida and pars tensa. The STAM system, which is used for staging cholesteatoma, divides the middle ear cavity into four sites: difficult access sites (S), tympanic cavity (T), attic (A), and mastoid (M). The difficult access sites (S) include S1, the supratubal recess (also called the anterior epitympanum or protympanum), S2, and the sinus tympani. Stage I indicates that the cholesteatoma is localized in the primary site—the attic (A) for the pars flaccida group and the tympanic cavity (T) for the pars tensa group. Stage II indicates that the cholesteatoma involves two or more sites. Stage III indicates the findings of extracranial complications or pathologic conditions including facial palsy, labyrinthine fistula, labyrinthitis, postauricular abscess or fistula, zygomatic abscess, neck abscess, canal wall destruction (more than half the length of the bony ear canal), tegmen destruction, and adhesive otitis. Stage IV indicates the presence of cholesteatoma along with intracranial complications including bacterial meningitis, epidural abscess, subdural abscess, cerebral abscess, sinus thrombosis, and brain herniation.

Table 1 Demographics and characteristics of the patients.a

Variables	n = 66	
Age, years	46.8 ± 13.9 (18–74)	
Gender		
Female	36 (54.5)	
Male	30 (45.5)	
Side of ear		
Left	33 (50.0)	
Right	33 (50.0)	
Follow-up periods, months	49.2 ± 21.6 (36–114)	
Preoperative PTA		
≤40 dB	20 (30.3)	
>40 dB	46 (69.7)	
Type of tympanoplasty		
Type I	20 (30.3)	
Type III-m	37 (56.1)	
Type III-M	7 (10.6)	
Type IV	2 (3.0)	
Group of cholesteatoma		
Pars flaccida	59 (89.4)	
Pars tensa/combined	7 (10.6)	
Stage of cholesteatoma		
Stage I	29 (43.9)	
Stage II–IV	37 (56.1)	
Ossicles destruction		
Normal or destruction of 1 ossicle	27 (40.9)	
Destruction of 2 ossicles	28 (42.4)	
Destruction of 3 ossicles	11 (16.7)	
Stapes		
Present	57 (86.4)	
Absence of suprastructure	9 (13.6)	
Sate of contralateral ear		
Normal	40 (60.6)	
OME/atelectasis/perforation	26 (39.4)	
Infection at the time of surgery		
No otorrhea	50 (75.8)	
Otorrhea	16 (24.2)	
Notes.

m minor columella

M major columella

OME Otitis media with effusion

a All values are presented as mean ± standard deviation with ranges in parentheses or numbers with percentages in parentheses.

Audiometric data were recorded at the time of admission before treatment and every 3 months after surgery in the first year and then every 6 months in the second year. Patients who did not undergo follow-up hearing tests for more than 3 years were excluded. During the treatment, hearing changes were examined by comparing hearing test results obtained before the treatment with those obtained at the latest follow-up. Pure-tone thresholds for air and bone conduction were conducted at the frequencies of 0.25, 0.5, 1, 2, 4, and 8 kHz. The pure-tone average (PTA) was determined by calculating the mean of the 0.5, 1, 2, and 4 kHz thresholds of air conduction. Air–bone gaps (ABGs) were calculated on the basis of air and bone conduction results obtained at the thresholds of 0.5, 1, 2, and 4 kHz. The mean preoperative and postoperative PTA and ABGs as well as improvements in the PTA and ABGs were recorded.

In this study, meeting the following criteria indicated successful hearing preservation after functional tympanomastoidectomy: (1) no recurrent or residual cholesteatoma during the follow-up period and (2) gain of at least 15 dB in the PTA when compared with preoperative data or conservation of the hearing value in patients with normal preoperative hearing (PTA ≤ 25 dB). Cholesteatoma recidivism was suspected via otoscope examination and deteriorated audiometry during follow-up period, and was established following revised surgery. The success rate of the hearing conservation was analyzed using the criteria suggested by the Japan Clinical Otology Committee (Sasaki et al., 2007). According to these criteria, 33 and 33 patients were included in the successful hearing group and unsuccessful hearing group, respectively.

To determine the factors that predict successful hearing preservation, the following variables were analyzed: age at surgery (≤45 years or >45 years), sex (female or male), preoperative PTA (≤40 dB or >40 dB), cholesteatoma groups (pars flaccida, pars tensa, or a combination of pars flaccida and pars tensa), cholesteatoma stage (stage I or stage II–IV), active infection at the time of surgery (absence or presence of otorrhea), ossicular chain destruction (normal or destruction of one, two, or three ossicles), stapes condition (presence or absence of the suprastructure), and state of the contralateral ear (normal or otitis media with effusion/atelectasis/eardrum perforation). One study evaluated the influence of the age factor on the prognosis for tympanoplasty revealed that the group aged under 45 years had a trend for higher hearing success, so we decided on 45 years as a criterion for two groups of patients (Tai, Ho & Juan, 1998). We put both patients with intact ossicular chain and with one ossicle destroyed in one group based on the Cholesteatoma-Atelectasis-Ossicle staging system of cholesteatoma (Kuo et al., 2012).

Functional tympanomastoidectomy was performed similar to the procedure reported in previous studies (Dornhoffer, 2000; Kuo et al., 2015). Under general anesthesia, a postauricular incision was made to expose the temporalis fascia, which was used for grafting. An adequate amount of the temporalis fascia was harvested to perform eardrum and external auditory canal (EAC) reconstruction later, including tympanoplasty and the coverage of bone grafts used for EAC reconstruction. A periosteal incision was made near the opening of the EAC but without penetration into the EAC lumen, followed by the elevation of the periosteum into the lateral ear canal. After identifying Henle’s spine, elevation of the posterior meatal epithelium continued until reaching one cm deep relative to Henle’s spine. A U-shaped incision was made over the elevated posterior meatal wall, and the vascular strip was then turned outside. A rubber Penrose drain was inserted to retract the auricle and lateral canal anteriorly. Weitlaner and Fisch articulated retractors (Karl Storz, Tuttlingen, Germany) were applied to gain further exposure. Bone grafts were harvested from the healthy mastoid cortex using a mastoid chisel and metal mallet (Karl Storz, Tuttlingen, Germany) and soaked in ofloxacin solution (Fig. 2A). After denuding around the perforation (if present), the tympanomeatal flap was raised. Canaloplasty and scutum removal were performed to gain access to the cholesteatoma. The extent of functional tympanomastoidectomy was dependent on how far the cholesteatoma had expanded, variously requiring atticotomy, atticoantrotomy, or atticoantromastoidectomy (Fig. 2B). After complete removal of the cholesteatoma epithelium, the open mastoid cavity, aditus ad antrum, and epitympanum were filled with autologous bone grafts for reconstructing and maintaining the anatomy of the EAC (Figs. 2C–2D). Temporalis fascia grafts were applied in an underlay fashion to cover the bone grafts and patch any eardrum defect. A superiorly based temporalis muscle flap was used to prevent cavity problems if the harvested bone grafts and fascial graft were inadequate. Tegmen dehiscence with dura exposure was covered and repaired using a pedicled flap. A smooth surface without any defects over the entire EAC was ensured after reconstruction.

Figure 2 Intraoperative views.

(A) Autologous bone grafts were harvested from the healthy mastoid cortex using a mastoid chisel. (B)Atticoantromastoidectomy was performed via inside-out approach to expose the cholesteatoma. (C) Type III tympanoplasty using cortical bone as columella was conducted in this case. (D) The open mastoid cavity, aditus ad antrum, and epitympanum were filled with bone grafts for reconstruction.

Tympanoplasty was performed as a single-stage procedure in all cases after the removal of cholesteatoma by using functional tympanomastoidectomy. Patients in this study underwent three types of tympanoplasty: type I, type III (Figs. 2C–2D), and type IV. Ossiculoplasty was performed in patients whose ossicular chain was destroyed by the cholesteatoma or removed for better exposure. Type III tympanoplasties performed in patients could be subdivided according to the ossiculoplasty technique used: (1) minor columella subtype (type III-m): the autologous incus, malleus, or cortical bone were harvested and sculptured as the columella between the stapes head and tympanic membrane and (2) major columella subtype (type III-M): an autologous bone graft or prosthesis was set between the stapes footplate and tympanic membrane. In type IV tympanoplasty, the fascia graft was set upon the mobile stapes footplate with a shielded round window (Merkus et al., 2018).

Statistical analyses were performed using SPSS version 24.0 (IBM Corp., Armonk, NY, USA). A paired-samples t test was performed to compare numerical variables and determine significant differences between preoperative and postoperative hearing results. One-way analysis of variance and Fisher’s exact test were used to compare hearing gains and the rate of successful hearing preservation, respectively, among different tympanoplasty types. Univariate and multivariate logistic regression analyses were performed to determine factors that were related to long-term hearing success. A p value of <0.05 was considered statistically significant.

Results

The mean age of patients was 46.8 ± 13.9 years (range: 18–74 years). Among 66 patients (66 ears), 28 were aged <45 years. Of 66 patients, 30 (45.5%) were men. The left and right ears were involved in 33 and 33 patients, respectively. Among 66 patients, 46 (69.7%) presented with more than mild hearing loss preoperatively (PTA >40 dB). The mean follow-up period was 49.2 ± 21.6 months (range: 36–114 months). Type I tympanoplasty was conducted in 20 (30.3%) of 66 ears. Type III tympanoplasty with the minor columella and type III tympanoplasty with the major columella were performed in 37 (56.1%) of 66 ears and 7 (10.6%) of 66 ears, respectively. Type IV tympanoplasty was performed in 2 (3%) of 66 ears. All patients were classified as pars flaccida, pars tensa, or a combination of pars flaccida and pars tensa according to tympanic membrane status. Of the 66 ears with cholesteatoma, 59 (89.4%), 5 (7.6%), and 2 (3.0%) were included in the groups of pars flaccida, pars tensa, and a combination of both pars flaccida and pars tensa, respectively. In this study, cases of pars tensa and the combined type were classified into the pars tensa group. A total of 29 (43.9%), 33 (50.0%), 4 (6.1%), and 0 (0%) patients had stage I, stage II, stage III, and stage IV disease, respectively. Of four patients with stage III disease, three had labyrinthine fistulas and one had delayed facial palsy. The contralateral ear was normal in 40 (60.6%) of 66 patients, and otitis media with effusion/atelectasis/ eardrum perforation was found in 26 (39.4%) of 66 patients. The ossicular chain status was confirmed surgically. A total of 27 (40.9%), 28 (42.4%), and 11 (16.7%) patients had normal ossicles or destruction in one ossicle, destruction in two ossicles, and destruction in three ossicles, respectively. Furthermore, 16 (24.2%) patients presented with active ear infection at the time of surgery, manifesting as otorrhea. Destruction of the stapes suprastructure was observed in 9 (13.6%) of 66 patients, and remained stapes integrity was noted in 57 (86.4%) patients. According to the criteria used to determine successful hearing preservation in this study, 33 (50.0%) patients were included in the successful hearing group and 33 (50.0%) patients were included in the unsuccessful hearing group.

Long-term hearing outcomes in patients who underwent surgery were evaluated by comparing the preoperative PTA and ABGs with postoperative results. The PTA (mean ± standard deviation [SD]) significantly improved from 50.78 ± 19.98 dB hearing level (HL) to 40.81 ± 21.22 dB HL (mean difference, 9.96 ± 13.73 dB HL; p < 0.001). ABGs (mean ± SD) significantly improved from 26.26 ± 10.53 dB HL to 17.58 ± 8.21 dB HL (mean difference: 8.48 ± 10.27 dB HL; p < 0.001; Table 2). The improvement in both PTA and ABGs demonstrated significant overall hearing recovery after functional mastoidectomy in patients with cholesteatoma.

Table 2 Surgical outcomes.a

Audiometric results	n = 66	p value	
Preoperative PTA (dB HL)	50.78 ± 19.98	<0.001*	
Postoperative PTA (dB HL)	40.81 ± 21.22	
PTA improvement (dB HL)	9.96 ± 13.73	
Preoperative ABG (dB HL)	26.26 ± 10.53	<0.001*	
Postoperative ABG (dB HL)	17.58 ± 8.21	
ABG improvement (dB HL)	8.48 ± 10.27	
Notes.

PTA pure-tone average

ABG air-bone gap

a All values are presented as mean ± standard deviation.

* p < 0.05 is considered statistically significant.

The hearing outcomes of different tympanoplasty types were evaluated on the basis of the hearing gains of the PTA and ABGs. Improvements in the PTA (mean ± SD) were 9.06 ± 11.52, 9.86 ± 13.35, 13.39 ± 23.03, and 8.75 ± 13.73 dB HL in type I, type III-m, type III-M, and type IV tympanoplasties, respectively. No significant differences in the improvement of the PTA were observed among different tympanoplasty types (p = 0.914). Postoperative gains in ABGs (mean ± SD) were 6.44 ± 9.86, 9.43 ± 10.41, 10.54 ± 12.46, and 4.38 ± 0.88 dB HL in type I, type III-m, type III-M, and type IV tympanoplasties, respectively. No significant differences in postoperative gains in ABGs were noted among different tympanoplasty types (p = 0.646). The preoperative PTA was compared between different tympanoplasty types to validate the reliability of the aforementioned results. A statistically significant difference in the baseline PTA was noted (p = 0.003), and the post hoc test results indicated that patients who received type III-M tympanoplasty had poorer preoperative hearing performance than did those who received type I tympanoplasty. The rates of successful hearing preservation were 55.0%, 51.4%, 28.6%, and 50.0% in type I, type III-m, type III-M, and type IV tympanoplasties, respectively. No significant differences in the rate of successful hearing preservation were observed between different tympanoplasty types (p = 0.788; Table 3).

Table 3 Hearing gains and successful hearing preservation in different types of tympanoplasties.a

	Type I	Type III-m	Type III-M	Type IV	p value*	
	(n = 20)	(n = 37)	(n = 7)	(n = 2)			
PTA gain	9.06 ± 11.52	9.86 ± 13.35	13.39 ± 23.03	8.75 ± 13.73	0.914	
ABG gain	6.44 ± 9.86	9.43 ± 10.41	10.54 ± 12.46	4.38 ± 0.88	0.646	
Preoperative PTA	39.88 ± 16.43	52.47 ± 19.92	66.96 ± 14.65	71.88 ± 6.19	0.003	Post hoc:
Type III-M >Type I	
Successful hearing preservation	11 (55.0)	19 (51.4)	2 (28.6)	1 (50.0)	0.788	
Notes.

PTA pure-tone average

ABG air-bone gap

Type III-m minor columella

Type III-M major columella

a All values are presented as mean ± standard deviation with ranges in parentheses or numbers with percentages in parentheses.

* one-way ANOVA test for comparison of hearing gain and preoperative PTA; Fisher’s exact test for rate of successful hearing preservation; p < 0.05 is considered statistically significant.

Logistic regression analysis was performed to predict factors that are related to long-term hearing success, and the results are listed in Table 4. The results of univariate analysis demonstrated that the likelihood of successful hearing preservation was significantly related to the early stage of cholesteatoma (odds ratio [OR] = 3.12; 95% confidence interval CI [1.13–8.60]; p = 0.028). There was a trend toward higher probability of successful hearing preservation in patients with pars flaccida cholesteatoma (OR = 7.11, 95% CI [0.81–62.79], p = 0.078) and those without otorrhea at the time of surgery (OR = 2.80, 95% CI [0.85–9.26], p = 0.091). Hearing success was not predicted by age (p = 0.619), sex (p = 0.324), preoperative PTA (p = 0.593), ossicular chain destruction (p = 0.389), integrity of the stapes (p = 0.290), and state of the contralateral ear (p = .315). The findings of multivariate logistic regression indicated that early-stage disease (OR = 4.41, 95% CI [1.26–15.49], p = 0.021) and pars flaccida cholesteatoma (OR = 13.55, 95% CI [1.19–154.68], p = 0.036) were significantly correlated with successful hearing preservation. Other variables were not found to be related to hearing success in multivariate analysis.

Table 4 Logistic regression univariate and multivariate analyses for hearing success.

Predictors	Cases	Success	Univariate	Multivariate	
	(n = 66)	n (%)	OR (95% CI)	p value	OR (95% CI)	p value	
Age							
≤45 years old	28	13 (46.4)	0.78 (0.29–2.08)	0.619	0.85 (0.24–3.06)	0.808	
>45 years old	38	20 (51.3)	1 (ref)		1 (ref)		
Gender							
Female	36	20 (55.6)	1.64 (0.62–4.34)	0.324	1.13 (0.33–3.87)	0.844	
Male	30	13 (43.3)	1 (ref)		1 (ref)		
Preoperative PTA							
≤40 dB	20	11 (55.0)	1.33 (0.47–3.83)	0.593	1.05 (0.25–4.46)	0.943	
>40 dB	46	22 (47.8)	1 (ref)		1 (ref)		
Group of cholesteatoma							
Pars flaccida	59	32 (54.2)	7.11 (0.81–62.79)	0.078	13.55 (1.19–154.68)	0.036*	
Pars tensa/ combined	7	1 (14.3)	1 (ref)		1 (ref)		
Stage of cholesteatoma							
Stage I	29	19 (65.5)	3.12 (1.13–8.60)	0.028*	4.41 (1.26–15.49)	0.021*	
Stage II–IV	37	14 (37.8)	1 (ref)		1 (ref)		
Infection at the time of surgery							
No otorrhea	50	28 (56.0)	2.80 (0.85–9.26)	0.091	2.92 (0.72–11.89)	0.135	
Otorrhea	16	5 (31.3)	1 (ref)		1 (ref)		
Ossicular chain							
Normal or destruction of 1 ossicle	27	14 (51.9)	1.89 (0.45–7.97)	0.389	1.01 (0.08–12.17)	0.993	
Destruction of 2 ossicles	28	15 (53.6)	2.02 (0.48–8.49)	0.337	1.03 (0.09–12.10)	0.979	
Destruction of 3 ossicles	11	4 (36.4)	1 (ref)		1 (ref)		
Stapes							
Present	57	30 (52.6)	2.22 (0.51–9.76)	0.290	1.24 (0.09–16.52)	0.871	
Absence of suprastructure	9	3 (33.3)	1 (ref)				
Sate of contralateral ear							
Normal	40	22 (50.0)	1.67 (0.62–4.52)	0.315	1.05 (0.31–3.62)	0.933	
OME/atelectasis/perforation	26	11 (42.3)	1 (ref)		1 (ref)		
Notes.

PTA (AC) pure-tone average of air conduction threshold in 0.5k, 1k, 2k and 4k Hz

OME otitis media with effusion

* p < 0.05 is considered statistically significant.

The rate of postoperative complications was 13.6% (9 of 66 ears). Delayed facial palsy was noted in one patient (1.5%) on postoperative day 7, but this patient exhibited complete recovery 1 month postoperatively. Two (3%) patients developed dry eardrum perforation, and two (3%) patient developed otitis media with effusion during the follow-up period. Four patients (6%) experienced recidivism during the follow-up period: three patients (4.5%) developed recurrence, and one patient (1.5%) had residual disease. The sites of recurrence in the three patients were all in the epitympanum region. A residual keratin pearl in the external auditory canal (EAC) was found in one patient. All four patients who experienced recidivism received revised surgery and recovered well without any new adverse events. Accumulation of keratin debris and purulent discharge in EAC were frequently noted during preoperative otoscope evaluation (Fig. 3A). At 12 months after surgery, 48 (72.7%) and 18 (27.3%) patients exhibited a healed and dry EAC (Fig. 3B) and a healed EAC with accumulated ear wax, respectively. High-resolution computed tomography (CT) of temporal bone were performed in all patients for preoperative evaluation (Fig. 4A). CT were also completed in all patients more than three years postoperatively. Postoperative CT scans revealed that the autologous bone grafts use for EAC reconstruction is well maintained during long-term follow-up (Fig. 4B). Persistent otorrhea related to granulation formation, poor graft uptake, and otomycosis was noted in four (6.0%) patients. All patients with significant ear discharge responded well to a topical ear solution and in-office local treatment.

Figure 3 Preoperative and postoperative otoscope images.

(A) Preoperative otoscope image showed keratin accumulates within the attic retraction pocket. (B) At 12 months after surgery, healed and dry external auditory canal and intact eardrum can be noted.

Figure 4 Preoperative and postoperative computed tomography.

(A) Preoperative computed tomography showed scutum erosion and retraction pocket into the epitympanum. (B) Computed tomography taken 4 years after surgery found that the bone grafts use for EAC reconstruction is well maintained (arrow) with good aeration in the middle ear cavity.

Discussion

The results of this study revealed that patients with acquired cholesteatoma presented significant overall hearing recovery with satisfactory recidivism occurrence rates after undergoing functional tympanomastoidectomy involving the use of autologous bone grafts as the primary material for reconstruction.

The use of the inside-out technique with the removal of the posterior auditory canal wall for exposure and eradication of cholesteatoma, followed by the reconstruction of the open cavity, has been proposed in several studies (Roth et al., 2013; Dornhoffer, 2006; Kim et al., 2019; Skoulakis et al., 2019; Lee et al., 2017). However, a flawless strategy for the management of the canal wall defect after cholesteatoma removal is not yet available (Mendlovic et al., 2021)

In their retrospective study conducted in 2004, Dornhoffer evaluated the long-term results of retrograde mastoidectomy, followed by the reconstruction of the auditory canal wall defect using the cymba cartilage (Dornhoffer, 2004). The harvested cymba cartilage in the fitting curvature was placed in the grooves for reconstruction. However, appropriate fitting of the cymba cartilage is challenging with an absolute learning curve. Shewel & Abougabal (2020) reported the surgical outcomes of retrograde mastoidectomy with canal wall reconstruction that involved the use of autograft bone plates that were secured in appropriate positions by fitting the graft in created grooves. Glass ionomer cement and cartilage pieces were used to stabilize the bone graft. Nonetheless, fitting and fixation of the bone plate were still challenging.

To the best of our knowledge, few studies have examined the long-term treatment efficacy and safety of applying autologous bone grafts as the primary material for reconstructing tympanomastoid defects in functional cholesteatoma surgery. Our retrospective study results showed that functional mastoidectomy followed by canal wall reconstruction with bone grafts significantly improved hearing and led to a low recidivism rate. An ideal surgical procedure should be quick and simple. Using bone grafts harvested from the healthy mastoid cortex for EAC defect reconstruction and mastoid obliteration without additional materials can improve operational efficiency.

According to audiometric outcomes, the PTA and ABGs demonstrated statistically significant improvements. This finding confirmed that this type of operation could yield reliable surgical results with significant recovery and preservation of hearing. The hearing outcomes of patients included in this study were comparable to those in previous studies on cholesteatoma surgery with canal wall reconstruction. Kuo et al. (2015) reported the long-term hearing results and effects of mastoid obliteration/exclusion using bone chips as the sole material for reconstruction in patients with cholesteatoma receiving retrograde tympanomastoidectomy. Significant gains in both PTA (mean difference, 5.7 dB; p < 0.001) and ABGs (mean difference, 6.96 dB; p < 0.001) were noted. In their retrospective study, Haeusler et al. reported that 586 patients (604 ears) received inside-out cholesteatoma surgery with cartilaginous reconstruction of the canal wall (Roth & Haeusler, 2008). A significant improvement in hearing was reported with a mean ABG of <30 dB in 78% patients postoperatively, and ABGs improved by an average of 12 dB.

Different tympanoplasty types exerted no significant effect on postoperative hearing gains and successful hearing preservation rates in our study. Patients who underwent type IV tympanoplasty tended to demonstrate poor hearing recovery with lower PTA and ABG gains. However, only two patients received type IV tympanoplasty, and the small sample size might have increased the margin of error. The integrity of the stapes was considered to be a crucial factor that predicts postoperative hearing outcomes. An intact stapes structure with a mobile foot plate (type I, II, and III tympanoplasty) showed a significant correlation with postoperative ABGs of 30 dB or lower in more than 80% of patients in one study; by contrast, this correlation was observed in only 70% of patients who underwent type IV tympanoplasty (Roth & Haeusler, 2008). No significant difference was observed in hearing outcomes between type III-m and type III-M tympanoplasties in our study. A study comparing type III-m with partial ossicular replacement prosthesis (PORP) and type III-M tympanoplasty with total ossicular replacement prosthesis (TORP) reported that both types of reconstruction showed no difference in hearing outcomes (Dornhoffer, 2004). Hence, restoration of ossicular coupling using either the autograft columella, PORP, or TORP could lead to satisfactory hearing results postoperatively, in contrast to type IV tympanoplasty with solely acoustic coupling (Merchant et al., 1997).

Patients and surgeons should know possible predictive factors related to successful hearing preservation after surgery. Multiple variables were included in univariate and multivariate logistic regression analyses. Early-stage cholesteatoma and pars flaccida cholesteatoma were determined to be independent predictors of hearing success in the multivariate analysis. This finding is consistent with that of a study conducted in 2020 by Ardic et al. (2020). The authors evaluated the surgical outcomes of cholesteatoma surgery and correlated them with the classifications of the newly proposed EAONO/JOS staging system. A significant correlation was observed between the classification of cholesteatoma and hearing results, and patients with pars flaccida retraction pocket cholesteatoma were observed to have the most favorable hearing outcomes. Furthermore, in all cholesteatoma groups, higher-stage disease was related to poor hearing results. Surgical techniques in the study included transcanal atticotomy and CWD and CWU mastoidectomies. Fukuda et al. (2019) investigated prognostic factors for hearing outcomes in patients with pars flaccida cholesteatoma according to the EAONO/JOS system. They noted favorable hearing outcomes in patients with early-stage cholesteatoma. However, different techniques including CWD and CWU mastoidectomies were used in the study. This finding suggests that an inconsistent correlation might exist between various surgical methods using the same staging system. Our study utilized the EAONO/JOS staging system for patients who received only functional cholesteatoma surgery; thus, its findings can be useful and reliable in the same group of patients. In our study, the preoperative hearing level was not a significant predictor. This finding is not consistent with those of previous studies. Kuo et al. (2012) reported that preoperative serviceable hearing was not a significant favorable predictor in multivariate analysis. Chadha et al. (2006) reported that air-conduction (AC) thresholds were increased by preoperative AC thresholds in patients who underwent CWU and CWD surgery. Inconsistent results among studies might be attributable to different surgical techniques. The stapes integrity has been considered to be a crucial predictor of successful hearing outcomes (Roth & Haeusler, 2008; Kuo et al., 2012). An intact stapes with a mobile foot plate was related to a higher probability of hearing success. The small sample size of the group with an absent stapes suprastructure might have led to nonsignificant results in our study. Risk factors such as ossicular chain damage, active infection of the middle ear, and contralateral ear disease were significant predictors of postoperative hearing outcomes reported in previous studies (Blom et al., 2015; Zwierz et al., 2019; Salviz et al., 2015). However, our findings did not support these results. With the improvement of surgical techniques and perioperative care, these risk factors might have less effect on hearing success.

The rate of recidivism in this study was 6% (4 of 66 ears): three (4.5%) patients developed recurrence and one (1.5%) patient had residual disease. A recent meta-analysis reported that the recidivism rate ranged from 5% to 17% in patients who underwent CWD mastoidectomy and 9% to 70% in patients who underwent CWU mastoidectomy (Tomlin et al., 2013). According to our results, functional tympanomastoidectomy involving the use of bone grafts for reconstruction not only could achieve satisfactory hearing preservation but also resulted in a lower rate of recidivism that was comparable to that of CWD mastoidectomy. Dornhoffer (2004) reported that the recurrence and residual disease were 16% (8 of 50 ears) and 4% (2 of 50 ears), respectively, in patients receiving retrograde mastoidectomy with cartilaginous canal wall defect reconstruction. Comparable surgical outcomes were also noted in a study in which patients underwent similar surgical procedures as patients in this study: 2.9% (3 of 102) of patients developed recurrence and 1% (1 of 102) of patients had residual disease (Kuo et al., 2015). All three cases of recurrence in our study were in the epitympanum region. Obliteration of the epitympanum or epitympanoplasty has been suggested to prevent the formation of a retraction pocket and to reduce the recurrence rate (Liu et al., 2014). The occurrence of recurrent disease in our study may be related to the incomplete exposure of the cholesteatoma matrix or late necrosis of the bone graft.

The major limitation of this study is the lack of comparison groups. Second, the retrospective analysis performed in this study may be a potential source of selection bias. Third, making comparisons among studies is difficult when evaluating hearing-related prognostic factors because no standard definition of successful hearing preservation is available. In the future, prospective trials that incorporate a standard definition of hearing recovery and a unified cholesteatoma staging system are needed. Studies examining the long-term outcomes of functional tympanomastoidectomy using primary autologous bone grafts for reconstruction are warranted because recurrence rates are highly related to the length of follow-up.

Conclusion

Functional cholesteatoma surgery with the use of autologous bone grafts for reconstruction could significantly improve hearing outcomes with low recidivism rates. Early-stage disease and pars flaccida cholesteatoma were two independent positive prognostic factors for successful hearing preservation. The combined use of staging and classification may provide patients and surgeons with valuable information on hearing outcomes.

Supplemental Information

Supplemental Information 1 Raw data

All patients with cholesteatoma admitted for functional tympanomastoidectomy using only bone chips for reconstruction in a single tertiary hospital between 2010 and 2018.

Click here for additional data file.

Supplemental Information 2 Codebook

Click here for additional data file.

The authors would like to thank all colleagues of department of Otolaryngology Head and Neck Surgery in China Medical University Hospital who provided insight and expertise that greatly assisted the research. This manuscript was edited by Wallace Academic Editing.

Additional Information and Declarations

Competing Interests

Author Contributions

Human Ethics

Data Availability

The authors declare there are no competing interests.

Wei-Che Lan conceived and designed the experiments, performed the experiments, analyzed the data, prepared figures and/or tables, and approved the final draft.

Ching-Yuan Wang and Ming-Hsui Tsai performed the experiments, authored or reviewed drafts of the paper, and approved the final draft.

Chia-Der Lin conceived and designed the experiments, authored or reviewed drafts of the paper, and approved the final draft.

The following information was supplied relating to ethical approvals (i.e., approving body and any reference numbers):

This study was approved by the Institutional Review Board of China Medical University Hospital (project approval number CMUH110-REC3-004).

The following information was supplied regarding data availability:

The raw measurements are available in the Supplementary File.

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
