# Peer review of "Long-term follow-up of applying autologous bone grafts for reconstructing tympanomastoid defects in functional cholesteatoma surgery"

_PeerJ, doi:10.7717/peerj.12522_

## Round 0.1 · original submission · Major Revisions

The study provides a significant amount of data that could have important clinical implications for functional cholesteatoma surgery treatment, but there are some criticisms raised by reviewers 1 & 2 that should be addressed, please pay attention to their comments.

·

Basic reporting

First of all, I would like to congratulate the authors for assembling such an interesting article. The article is generally well-written and structurally adequate. The authors have successfully built the rationale behind their study, providing solid grounds to demonstrate why their study would be important to the literature and practicing ENT physicians.
I have minor general concerns at this point of the review:
- Although I commend the authors for performing such a detailed literature review, the manuscript is too long. Some of the informations contained in the introduction and discussion are of general knowledge of physicians who are more likely to read the manuscript. Therefore, I recommend shortening the manuscript to include only information that is critical to the understanding of the article.
- The english language is generally acceptable. However, some sentences are wordy, with minor syntax issues. I recommend the article should be proof-read by an expert.
- There are some sentences in the introduction and discussion that are lacking references.
- Another surgical technique proposed in the past that is similar to yours was also described by Prof. Paparella, with the name "flexible endaural approach". I think this is worth mentioning in the introduction.
- The images are excellent; however, they are very small and condensed in a single picture. I strongly suggest the authors to separate the figures in at least 3 different pictures so they can be expanded and seen in more detail. Additionally, it would be interesting to include more pictures of how the bone chips were harvested.

Experimental design

The experimental design was described in detail, and seems adequate to answer the proposed objectives. I particularly enjoyed the strictness of your inclusion/exclusion criteria, and the detailed description of the surgical technique. However, I have major concerns regarding the experimental design used:
- There is no control group to compare the findings.
- Were all patients operated by the same surgeon? If the surgeries were performed by more than 1 surgeon, it would be interesting to describe if the surgical technique changed among surgeons and if there were a significant heterogeneity in the results obtained by each one of them.
- The authors performed hearing tests at the 3-year timeline following the surgeries. Although this is very interesting, the lack of a control group does not allow comparison regarding the success of this specific reconstruction (using bone chips) with other approaches. As the auditory results are largely dependent of the degree of bone erosion/ossicular chain abnormalities, it seems that this information - although interesting - does not add significantly to your results of conclusions.
- In the same line of thought, it is not possible to compare the success regarding lack of recurrence as there is no control group to compare the findings.
- I was particularly confused by the use of the term "bone chips". Bone chips usually refer to particles of bone that accidentally separate from the drilled area. As the "methodology" only briefly cites how the "bone chips" were harvested and there are no pictures to illustrate this, I thought that the authors have collected bone pate and performed a mastoid occlusion with this material before I saw the surgical image later on. I would personally recommend the use of other terminology to better illustrate the technique used - such as "autologous bone grafts", or "bone graft palisade".
- There are plenty of articles that performed a bone reconstruction of the posterior canal wall using different techniques that were not cited or included in the discussion.

Validity of the findings

The results are meaningful and interesting. The data yields interesting information that could potentially improve the practice of ENT physicians that are involved with cholesteatoma surgery. Although the article could be a significant contribution to the literature, the lack of a control group critically undermines the validity of your results.

Reviewer 2 ·

Basic reporting

I would like to congratulate the authors, as this is a very well-written study, and although it is a retrospective study, methodologically, inclusion and exclusion criteria were carefully used, making the sample very homogeneous, making the results comparable with other studies. However, only 66 ears in total are a small sample as some subgroups were left with an extremely limited number of patients (less than 10 individuals) and some variables that are not very prevalent, such as the performance of Type III-M, Type IV tympanoplasty and absence of the stapes suprastructure, were not very representative and therefore are difficult to be statistically evaluated.
I must emphasize that both the figures and the tables are very clear and informative.
I suggest checking references number 20 and 27 as they are outside the formation suggested by PeerJ.

Experimental design

This manuscript presents a follow-up of at least 3 years with functional results within expectations both from the point of view of relapse, recurrence and hearing recovery, although in cases of cholesteatoma there is the possibility of late recurrences with a 5 year evolution. In relation to the surgical aspects used for the removal of the cholesteatoma, the material used for the reconstruction does not present any novelty and the impact can be evaluated only in relation to the follow-up time and in relation to the preservation of the auditory thresholds as the recovery rate observed despite the statistically significant was only 10dB, which may or may not be clinically relevant depending on the previous aero-bone gap of everyone.

Validity of the findings

Studies on cholesteatoma and its treatments have been carried out since the twentieth century and unfortunately, the surgical exeresis of cholesteatoma, regardless of its origin, is still the only therapy capable of controlling the evolution of this disease and preventing its complications in both the auditory aspect and its fatal risk. Several techniques are described and currently the most used are those that remove the disease trying to preserve the anatomical structures responsible for the auditory function, a technique used in the study and called functional surgery for cholesteatoma. The reconstructive phase of this surgery has also been widely explored and several materials have been reported with the individual's own cortical bone but also several other synthetic materials for the reconstruction of both the posterior wall and the compromised ossicular chain. Therefore, neither the surgical technique used, nor the material used for reconstruction are innovative, on the contrary, it has been widely used because economically it does not generate additional costs with heterologous prostheses, and when performed by well-trained surgeons, they achieve reasonable functional results. For this reason, there are already several studies that examine the results of the follow-up of the application of autograft bone to reconstruct tympanomastoid defects in surgery for functional cholesteatoma.
The conclusions are based on the results; however, it does not add any additional data to those already provided by the current literature.

Annotated reviews are not available for download in order to protect the identity of reviewers who chose to remain anonymous.

·

Basic reporting

No comments

Experimental design

No comments

Validity of the findings

No comments

Additional comments

No comments

---

## Round 0.2 · accepted · Accept

All comments have been addressed

·

Basic reporting

No issues found

Experimental design

No issues found

Validity of the findings

Although the study replicates methodology from other past studies, the findings are valid.

Additional comments

Thank you for addressing the comments made by the reviewers. The manuscript has increased a lot in quality. I have no further comments.

Reviewer 2 ·

Basic reporting

Thank you for addressing the comments in reviewing this manuscript. The authors provided additional information and after had been proof-read by an expert and adopting specific terminology, the manuscript improved overall. The findings are interesting and remain relevant to the journal's readers.

Experimental design

Although it was not possible to expand the sample size in different subgroups, the results indicate an adequate functional result using autologous bone grafts for reconstruction external ear canal defect and mastoid obliteration without additional materials can improve operational efficiency, which greatly reduces surgical costs.

Validity of the findings

The findings are interesting and remain relevant to the journal's readers.

Additional comments

The separation of the figures really facilitated the interpretation of the data however in figure 1, the word "Cholesteatoma" has a typo and must be corrected. I also suggest putting an acronym for CMO, CWD, EAC.

Annotated reviews are not available for download in order to protect the identity of reviewers who chose to remain anonymous.